# Analyzing Low-Level mtDNA Heteroplasmy—Pitfalls and Challenges from Bench to Benchmarking

**DOI:** 10.3390/ijms22020935

**Published:** 2021-01-19

**Authors:** Federica Fazzini, Liane Fendt, Sebastian Schönherr, Lukas Forer, Bernd Schöpf, Gertraud Streiter, Jamie Lee Losso, Anita Kloss-Brandstätter, Florian Kronenberg, Hansi Weissensteiner

**Affiliations:** 1Department of Genetics and Pharmacology, Institute of Genetic Epidemiology, Medical University of Innsbruck, A-6020 Innsbruck, Austria; federica.fazzini@gmail.com (F.F.); liane.fendt@i-med.ac.at (L.F.); sebastian.schoenherr@i-med.ac.at (S.S.); lukas.forer@i-med.ac.at (L.F.); bernd.schoepf@gmail.com (B.S.); gertraud.streiter@i-med.ac.at (G.S.); jamie_lee1515@hotmail.com (J.L.L.); A.Kloss-Brandstaetter@fh-kaernten.at (A.K.-B.); Florian.Kronenberg@i-med.ac.at (F.K.); 2Carinthia University of Applied Sciences, A-9524 Villach, Austria

**Keywords:** mitochondrial DNA (mtDNA), next generation sequencing (NGS), heteroplasmy, variant callers, DNA polymerase

## Abstract

Massive parallel sequencing technologies are promising a highly sensitive detection of low-level mutations, especially in mitochondrial DNA (mtDNA) studies. However, processes from DNA extraction and library construction to bioinformatic analysis include several varying tasks. Further, there is no validated recommendation for the comprehensive procedure. In this study, we examined potential pitfalls on the sequencing results based on two-person mtDNA mixtures. Therefore, we compared three DNA polymerases, six different variant callers in five mixtures between 50% and 0.5% variant allele frequencies generated with two different amplification protocols. In total, 48 samples were sequenced on Illumina MiSeq. Low-level variant calling at the 1% variant level and below was performed by comparing trimming and PCR duplicate removal as well as six different variant callers. The results indicate that sensitivity, specificity, and precision highly depend on the investigated polymerase but also vary based on the analysis tools. Our data highlight the advantage of prior standardization and validation of the individual laboratory setup with a DNA mixture model. Finally, we provide an artificial heteroplasmy benchmark dataset that can help improve somatic variant callers or pipelines, which may be of great interest for research related to cancer and aging.

## 1. Introduction

One of the most precious benefits of massive parallel sequencing (MPS) technologies in the field of mitochondrial DNA (mtDNA) research is the increase in sensitivity for detecting heteroplasmy, a state where at least two different mtDNA molecules are present in one mitochondrion, cell, or tissue. While the detection level is limited to approximately 10% with former gold standard Sanger-type sequencing [1,2], the abundance of resulting sequencing reads from a single run derived from MPS platforms now allows for a much more precise estimation of minor variant allele frequencies. For the time being, the most widely used sequencing technology stems from Illumina. The method is based on the sequencing-by-synthesis approach. Illumina uses fluorescently labeled nucleotides to sequence library clusters on a flow cell surface. Incorporated deoxynucleoside triphosphates (dNTPs) are visualized by an optics-based technology identifying the signal intensity of the dye.

In population genetics, low-level (<10%) heteroplasmy information is valuable for the estimation of mutation rates, the calibration of the genetic clock [3], and for the estimation of purifying selection in a population [4,5]. In forensic genetics, the discernibility of two sequence profiles is essentially elevated if heteroplasmy can be taken into account [6,7]. The detection of low-level variants is also of great importance in medical genetics for diagnosis and prognosis of a mitochondrial disease [8] but also in cancer research [9,10,11,12] or aging-related research [13]. A study of Ye et al. [14] reporting extensive pathogenicity of mitochondrial heteroplasmy in healthy human individuals was the starting point of a discussion about published sequence quality derived from MPS platforms [15,16]. The group investigated 1085 human mtDNA sequences from the publicly available 1000 Genomes Project dataset. They reported that 90% of the individuals harbored at least one (low-level) heteroplasmy and that at least 20% of those heteroplasmic variants had been connected to diseases. However, a precise inspection of the heteroplasmy data revealed that contamination by sample mixture (carryover) was the source of at least some of the apparent low-level variants [15]. Moreover, Just et al. generally questioned the ability of the current MPS technologies and bioinformatics approaches to reliably detect levels of heteroplasmy lower than 5–10%, and hence the outcome of large scale sequencing studies based on current sequencing approaches [17]. This claim has to be taken very seriously since the entire scientific mtDNA field has already begun to report and interpret analysis results cutting slowly the proclaimed 1% threshold [18,19].

Experimental factors are known to heavily influence error rates in Next-generation sequencing (NGS) datasets. A recent report has shown that both library preparation method and sequencing chemistry can have an impact on the error profiles of sequencing data generated on Illumina platforms [20]. For example, the use of transposome-based library preparation techniques (e.g., NexteraXT) has been associated with specific sequencing biases, while the use of long-read sequencing chemistries (e.g., MiSeq) is known to produce higher error rates. The use of alternative amplification strategies significantly impacts heteroplasmy detection, which can lead to potentially erroneous results [21]. With the advent of more sensitive sequencing techniques like NGS, the detection and reporting of potential artefacts created due to improper amplification strategies are likely to grow more prominently in the future. For this reason, we decided to re-evaluate the impact of amplification strategies in the era of NGS by examining the influence of three widely used polymerase enzymes on heteroplasmy detection and potential amplification artefact profiles.

The aim of this study was to determine the sensitivity, specificity, as well as the precision to detect minor variants within MPS data derived from the Illumina MiSeq platform by applying an mtDNA mixture model (i.e., pairwise mix-ups) of a defined sequence pattern. First, we investigated whether pre-analytical factors such as the use of different DNA polymerases for the library preparation process could potentially bias the sequencing results. Subsequently, we explored the impact of preprocessing the sequencing data by applying read trimming and duplication removal on the resulting raw data. Finally, we assessed several variant calling tools on our benchmarking dataset in order to investigate the post-processing for the mtDNA heteroplasmy analysis.

## 2. Results

### 2.1. Effect of Starting Material and DNA Polymerases

Samples for the mixture experiments were prepared in two different ways (Figure 1a). First, the DNA of two individuals was mixed at five different ratios (M1 (1:2), M2 (1:10), M3 (1:50), M4 (1:100), M5 (1:200)), which then was amplified by three different enzymes (called “total DNA extracts”). Second, the PCR products, derived from the same two DNAs, were mixed at the ratios as described above (called “PCR products”). All variants that were not shared between both haplotypes emerged as apparent heteroplasmic variants in the mixture types (Figure 1b, blue and green font).

First, we evaluated the potential impact of the starting material (total DNA extracts vs. PCR products) on the precision for low-level (at the 1% level) polymorphic variant detection in the different mixtures. All detected mixture levels averaged over all expected sites of the minor component (variants in Figure 1b blue font) were in agreement with the expected levels for PCR products and for total DNA extracts as well as for the three different polymerases (see Figure 2, Appendix A).

As expected, the PCR products showed variant levels closer to the expected mixture, compared to the corresponding total DNA extracts. Furthermore, for both mixture types a position-wise comparison of the mixture levels revealed deviations from the expected mixture values depending on the genetic loci. Mixture ratios identified at positions *16234*, *16256*, and *16270* revealed the lowest values (e.g., considering M3 a mean of 12.5% decrease in the DNA extracts and 39.5% in PCR products), whereas highest values were detected at positions *12372*, *3197*, and *7028* (e.g., for M3 a mean of 19% increase in the DNA extracts and 9% in PCR products) over all of the experiments.

Second, we investigated the performance of three different Taq polymerases: Clontech LA Advantage (Clontech/CLAA), LongAmp Taq Polymerase (NEB), and Herculase II Fusion (HERK). For this, the results of M1, M2, M3, and M4 between both library types of the mixture model were compared with the default settings by applying the variant caller mutserve, the updated version of mtDNA-Server [22]. As the threshold for variant detection is 1%, the analysis of the M5 mix-up (0.5% variant level) was rendered obsolete.

The comparison with our gold standard allowed calculation of false positive and false negative Single-nucleotide polymorphisms (SNPs) detected by each of the three polymerases. The gold standard was defined based on the sequencing of the unmixed source samples and additionally compared to the previous mixture model on the Illumina HiSeq (see Materials and Methods section and Data Availability subsection). The number of false positive variants was much higher in Clontech (mean = 15 per sample) and NEB (mean = 9.2 per sample) samples as compared to HERK (mean = 0.3 per sample) in all the mixtures. In this analysis, the data was run using mutserve’s default parameters (1% variant allele threshold, per base quality Phred Score 20; however, the per Base Alignment Quality (BAQ) [23] turned off—see Appendix A for more details) without further preprocessing steps. Therefore, HERK displayed the best performance (Figure 3 and Figure 4, Appendix A). The mean variant level of false positives was 0.77% for Clontech, 0.70% for HERK, and 1.38% for NEB (Appendix A).

### 2.2. Impact of Read Trimming and Removing Duplicate Reads

In a next step, we investigated the impact of bioinformatic quality control (QC) steps on the data quality of mitochondrial genome sequences by performing adapter trimming and quality filtering, as well as duplication removal. As part of this assessment, we compared the mixture model with and without applying fastp, an all-in-one FASTQ preprocessor [24], as well as Picard-Tools MarkDuplicates (http://broadinstitute.github.io/picard). The mean coverage over all mixtures M1–M4 dropped by 3% when applying fastp and by 14.6% when removing duplicate reads, compared to the untreated FASTQ files (mean coverage ~8900×). Where possible, the per base alignment quality [23] was turned off. The overall performance comparison by estimating the sensitivity, specificity, and precision showed only minor differences between the trimmed data, the de-duplicated data, and the untreated data, as shown by the F1 score (Figure 3). Interestingly, no additional benefit for the FASTQ preprocessing steps (fastp, markdup in Figure 3) based on the present MiSeq mixture model could be observed.

### 2.3. Performance of Different Variant Callers

As we put special emphasis on reliably detecting low-level mutations below the 1% threshold, we assessed mixtures of 1:100 and 1:200 (M4 and M5, corresponding to a 1% and 0.5% minor allele frequency, respectively; Figure 1a). As the preprocessing step indicated no advantage for trimming or de-duplication of the reads, we applied the variant caller onto the Binary Alignment Map (BAM) files without preprocessing. A per-base quality of Q30 (indicating a 0.1% error per base) was used for all tools (where this parameter could be indicated), resulting in a median coverage over all M4 and M5 mixtures of ~5900× but differing slightly between the samples (Appendix A). In order to calculate sensitivity, specificity, and performance of the variant callers, all the M4 and M5 samples were analyzed with Freebayes [25], GATK Mutect2 [26,27], LoFreq [28], mutserve [22], VarDict [29], and VarScan 2 [30] (Appendix A). Previously, Stead et al. [31] found VarScan 2 showing best performance for identification of low-allele fraction variants down to 1% minor allele frequency. Table 1 gives an overview of the applied variant caller with the parameters, the used version, and the mean runtime per sample. More detailed information about the variant callers can be found in Appendix A.

There was a significant difference (one-way ANOVA, *p* = 0.00095) in the average number of false positive low-level variants among samples processed with different polymerases, ranging from a mean of 997 for the Clontech and 868 for the NEB polymerase down to 47 for HERK when comparing results from all variant callers over the two different extraction methods. On the one hand, analysis based on the polymerase revealed that using NEB, heteroplasmies were detected at highest sensitivity over all experiments when compared to the enzymes HERK and Clontech. On the other hand, HERK showed the highest number of false negatives (i.e., expected variants that could not be found; however, no significant difference between false negatives (one-way ANOVA, *p* = 0.36), see Appendix A for details). Those samples starting analysis from PCR products showed better performance over the samples derived from total DNA mixtures.

Interestingly, all samples amplified with the NEB enzyme produced a phantom mutation [32] at position *3210* (*C > A*) and was detected by each variant callers but was absent in the source files of H1 and U5. The artefact was present in all NEB PCR mixtures at an average of 6.5% and in all total DNA NEB mixtures at an average 4.3%.

Our mixture model investigations revealed Freebayes, mutserve, and VarScan 2 analyses to display comparable results concerning the sensitivity for the 1% and 0.5% minor contribution detection. Sensitivity values retrieved from GATK Mutect2 analysis were generally lower for all mixture types. The mean specificity was above 93% for all the variant callers and also for the different enzymes.

Precision values for detection of the distinct mutational pattern varied strongly among the investigated analysis pipelines but even more among different polymerases. Due to an extraordinary high number of false positive low-level polymorphic positions exceeding the 0.4% level applied for most variant callers (not explicitly set in GATK4 Mutect2 and Lofreq), the highest precision values for pattern detection ranged from 3% to highest 6% maximum for NEB and Clontech using all the variant callers. Best values were retrieved from libraries based on HERK enzyme; an average of 56% precision was reached in total over all experiments (ranging from 7–8% with LoFreq and 94–97% using VarDict). Similarly, F1 scores (Figure 4) based on the sensitivity and precision were calculated. See Appendix A for the detailed information about the estimated performance parameters.

## 3. Discussion

The feasibility to precisely assess mtDNA low-level heteroplasmy in MPS data has important implications for a variety of scientific fields and has led to an ongoing discussion [17]. Indeed, the literature displays a diversity of articles indicating remarkable numbers of low-level sequence variants that are not expected given the current knowledge about the human mitochondrial phylogeny [15,33]. Interpretation of false positive associations is misleading in any particular case and might guide future research in wrong directions. Therefore, data QC, defining the limitations of the applied method and a technical evaluation, at its best is crucial to receive high quality data.

The rapid development, methodical adaptions, and improvements within the Illumina MiSeq system are raising new challenges. The technique is clearly outperforming conventional Sanger‑type sequencing by means of parallelization and the potential to quantify heteroplasmy given a homogeneous read coverage over the whole mitochondrial genome [1]. On the other hand, it is in the nature of novel approaches that no sufficient evaluation strategy exists by which unexpected single or systematic artefacts can be perceived, particularly if a workflow bears an extensive range for variability. By applying different long-range polymerases within the library construction process, we discovered the emergence of a systematic artefact at position *3210* present in any independent sequencing reaction at equal levels caused exclusively by the NEB enzyme. It is worth noting that the arguable enzyme was the cheapest in the observed group. In our previous work [12] this variant was described as heteroplasmic, but in the light of this observation it would probably be more appropriate to classify it as an artefact. This heteroplasmic site at position *3210* manifested itself in one benign tumor pair with heteroplasmy levels of 1.8% and 1.7%, respectively (Patient KT013), and one additional cancer sample (Patient KT012 with 6.2% heteroplasmy level) but not in the corresponding benign control. As we based our main findings on variants with a heteroplasmy level difference between tumor and paired benign samples of at least 20%, the presence of *3210* does not influence the final conclusion. Randomly introduced polymerase errors would be statistically eradicated by being filtered out based on the defined cut-off setting in the bioinformatic analysis. Systematic errors however, such as those identified in the present study are—if not questioned by a sophisticated evaluation procedure—prone to be reported as an apparent heteroplasmy following false interpretation in the clinical context in the worst case. The critical tuning of the mtDNA analysis pipeline favors the appropriate balance between sensitivity and specificity for variant calling. This can be seen for instance on the M4–M5 mixtures, where mutserve revealed the lowest effectiveness for detecting false positives in noisy data (NEB, Clontech), although it was also the best in avoiding false negatives.

Fluctuations were found to be systematic position-wise over the individual sequencing reactions rather than being randomly distributed over the whole mitochondrial genome. For instance, the lower variant levels on positions *16234*, *16256*, and *16270* could be the result of the alignment/mapping quality filter—as all 3 variants can be detected on one single read within 36 base pairs. This is not only an important finding in terms of the feasibility to precisely assess individual heteroplasmy levels but also in terms of the possibility to take advantage of the MPS reads to use for copy number detection. Estimation of mtDNA copy number relative to the nuclear copy number requires the homogeneous and unambiguous representation of reads at every position, and one has to take into account a potential bias in this regard.

The comparison between different variant calling tools and polymerases has been explored in nuclear DNA in several prior studies [34,35,36,37,38,39]. While different research groups investigated mtDNA mixtures on different MPS devices in recent years [40,41,42], only a few have evaluated them on mtDNA sequence data in greater detail below the 1% threshold [6,43]. In this study, we prepared haplotype mixtures of total DNA extracts as well as from mitochondrial PCR products at five different ratios down to 0.5% minor allele frequencies, with subsequent library construction. Amplification within either step was performed using three different polymerases (Clontech LA Advantage, Herculase II Fusion, LongAmp Taq Polymerase) resulting in 48 samples sequenced on Illumina MiSeq. The use of artificial mtDNA mixtures has also been published previously [42,44,45,46], but our investigation is the first to evaluate extensively the impact of the polymerases, read trimming, duplicate reads removal, and the assessment of current variant callers. In conclusion, our results demonstrate the vulnerability of a system in a per se robust workflow. We therefore strongly recommend evaluating the individual laboratory and bioinformatic pipeline for any mtDNA workflow that is normally applied on several experiments or even large-scale studies. Such an a priori validation could, for example, include the testing for the reliability of the used enzymes based on a mixture model, as demonstrated in the present investigation, which could even be expanded to additional alternative haplotype mixtures. Of course, in this regard, limitations will remain, such as the fact that all observations and irregularities are restricted to the particular haplotype model. Nevertheless, we proved that we were able to identify artefacts introduced in the laboratory process. In our case, we could obtain the best results with the HERK enzyme by using the mutserve variant caller, while we obtained the worst result when using the Clontech enzyme, where all variant callers had substantial issues (see Figure 4).

Further, we investigated the additional detected variants between the 0.5% and 1% level, not expected by the gold standard. Those apparent false positive variants that were present in the NEB and Clontech mixtures were mostly G-to-A (~34%) and T-to-C transitions (~65%). Appendix A gives an overview of the nucleotide distributions. The result is based on the mutserve data, which also underwent a strand bias check. This is an important aspect since sequencing errors are strongly biased to one orientation, while PCR errors can be observed for both forward and reverse strands.

For improving quality and increasing sensitivity for NGS libraries, good and sufficient starting material is key. Minimizing bias, using high fidelity Taq polymerase, and optimizing fragmentation conditions are further crucial steps. Additionally, standardization with appropriate quality management (QM), quality assessment (QA), and QC tools, as well as laboratory guidelines for NGS, are important instruments [47,48] for increasing data quality. Moreover, we highlight the advantage of prior standardization and validation of the individual laboratory setup with a DNA mixture model.

The data that were generated by this project are provided as a low-level variant calling benchmarking dataset and can be helpful in validating additional variant callers or novel approaches for both data pre- and post-processing. Thereby, not only mtDNA-related research can benefit but also disciplines where somatic DNA mutations of low-level frequencies are of interest, predominantly cancer and aging-related research. In this regard, it should be added that not all investigated variant callers within this work were developed explicitly for the detection of low-level heteroplasmy in mitochondrial sequencing studies. More details about the variant callers can be found in the Appendix A. The raw sequencing data of the source files as well as all artificial heteroplasmy mixtures together with the BAM files of the MiSeq runs as well as the HiSeq BAM files are available at Zenodo (see Data Availability subsection).

## 4. Materials and Methods

### 4.1. Statement/Samples

Peripheral blood was collected for earlier sequencing projects [1,49] approved by the Medical University of Innsbruck (study codes UN3564 and AN 4837 respectively) and sequenced as part of Fendt et al. [12] (Ref. No. AN2016-0026 359/4.2 366/5.8 (3923a)) while beeing carried out in accordance with the Code of Ethics of the World Medical Association (Declaration of Helsinki). Informed consent was obtained from the subjects; both were older than 18 years of age. The mtDNA sequences were deposited in GenBank with entries KC286589 and HM625679. Respectively, the haplotypes belonged to haplogroup H1c6 and U5a2e according to HaploGrep2 [50] based on Phylotree 17 [51] (Figure 1b). The raw data of the mixture model can be found on Zenodo (see Data Availability for details). Appendix A lists all samples with details about coverage and performance.

### 4.2. DNA Extraction and Mixture Preparation

DNA of the two reference individuals was extracted automatically using the Qiagen EZ1 advanced Biorobot (QIAGEN, Hilden, Germany) with the Qiagen EZ1 DNA blood kit following the manufacturers recommendations. Total DNA of each donor was quantified by spectrophotometry using Tecan NanoQuant Infinite M200 (Tecan Group Ltd., Männedorf, Switzerland). DNA mixtures were prepared by combining total DNA extracts in defined ratios: M1 (1:2), M2 (1:10), M3 (1:50), M4 (1:100), and M5 (1:200), being subsequently amplified. In parallel, PCR products were generated according to Fendt et al. [52] and Kloss-Brandstätter et al. [1] using three different enzymes: Clontech LA Advantage (TaKaRa Bio Inc., Kusatsu, Japan), Long Amp (New England Biolabs, Ipswich, MA, USA), and Herculase II Fusion (Agilent Technologies, Santa Clara, CA, USA). The different polymerases were all described to exhibit a proofreading activity. However, the error rates per base were slightly different (data reported by the companies’ websites): TaKaRa LA advantage (Clontech): 25 errors per 100,000 bp; LongAmp Taq polymerase (NEB): 1–20 errors per 100,000 bp; Herculase II fusion: 0.13 errors per 100,000 bp. After amplification and before purification, all PCR products were analyzed on an agarose gel to check for amplicon size and the presence of undesired, non-specific products. For this, 2 µL of each PCR product was evaluated on a 0.7% agarose gel containing GelRed (Biotium, Fremont, CA, USA) by comparing to a 1 kb ladder (Fermentas, Waltham, MA, USA) and visualized on a BioRad Gel Doc XR (BioRad, Hercules, CA, USA). An example of an agarose gel of samples amplified using the NEB Taq polymerase, as it was the one showing phantom mutations, can be found in the Appendix A. PCR products were purified using Agencourt Ampure XP beads (Beckman Coulter, Brea, CA, USA), quantified spectrophotometrically, and mixed as described above at 1:2, 1:10, 1:50, 1:100, as well as 1:200 ratios (Figure 1a).

### 4.3. Sanger and Illumina HiSeq Sequencing of the Reference Samples

To validate the MPS analysis we performed gold standard Sanger-type sequencing of the entire mtDNA of the two mixture component samples, as presented in Kloss-Brandstätter et al. [1]. In brief, whole mtDNA was amplified as two overlapping 8.5 kb fragments as described in Fendt et al. [12,52]. Libraries for the HiSeq 2500 of the unmixed DNA reference samples were prepared as described in Kloss-Brandstätter et al. [1].

### 4.4. Library Construction and Sequencing of Mixture Types on Illumina MiSeq

Libraries for the Illumina MiSeq of the DNA and PCR mixtures were constructed as presented in Fendt et al. [12] by preparing 500–800 bp inserts of each 200 ng of pooled fragment A and B. In a first step, the pooled samples were enzymatically fragmented in a final volume of 20 µL containing 2 µL of 2 × dsDNA Fragmentase Reaction Buffer v2 and 2 µL of 200 mM MgCl_2_ and 2 µL NEBNext dsDNA Fragmentase (New England Biolabs, Ipswich, MA, USA) by incubation for approximately 6 min at 37 °C. The resulting fragments were purified using MagSi-NGS^PREP^ beads (MagnaMedics Diagnostics, Geleen, The Netherlands). Dual-indexed libraries were produced using the TruSeq Nano DNA HT sample preparation kit (Illumina, San Diego, CA, USA) according to the manufacturer’s instruction with minor modifications: 47.5 µL and 15 µL of undiluted Sample Purification Beads (AMPure, Illumina, San Diego, CA, USA) were used for right and left side size selection, respectively. Furthermore, for final library enrichment, a total of 9 amplification cycles were performed. Size distribution of the enriched libraries was determined on the Fragment Analyzer system (Agilent, Santa Clara, CA, USA) using the DNF-930 dsDNA Reagent Kit (75–20,000 bp), and concentration was determined by fluorometric quantification using QuBit 3.0 (ThermoFisher, Waltham, MA, USA). Pooled libraries were clustered with a concentration of 12 pM on a standard v2 flow cell and sequenced using the MiSeq Reagent Kit v2 (500 cycles) to generate approximately 7–8 Gbp of data.

### 4.5. Gold Standard

Our reference combined sample was defined by 27 expected artificial heteroplasmic sites (result of mixture of the two haplotypes H1c6 and U5a2e in reference to the rCRS [53]), 6 homoplasmic variants, as well as one (in M1) and three (in M2, M3, M4, M5) private mutations. Those private mutations were identified within the Illumina Hiseq runs [1] and confirmed by sequencing the source samples with all enzymes on the MiSeq as well (data included). The false positive variants were defined as sites present in a minority of samples (<8) and amplified only with one or two polymerases out of three. The sites with inconsistency in the results (*n* = 10) were excluded from the analysis and treated as neutral. Appendix A lists all variants defining the gold standard, including variants that were not considered in the analysis due to reference issues, other issues (homopolymeric stretch or AC repeat), or where the variant levels were below 1%.

Appendix A lists all details of the six different variant calling tools, the used version, a download of the source codes or executable code, as well as the parameters applied for validation. Additional Appendix A reports all details of the 48 sequences, with the corresponding mixture level, information on enzyme, DNA extraction, as well as the F1 scores for all mutserve 1.3.4 runs over all mixtures M1–M4 with the different preprocessing steps. Furthermore, the 12 samples investigated for ultra-sensitive variant calling include the median F1 score over all variant callers in the mixtures M4 and M5 (1% and 0.5% variant level, respectively).

## Figures and Tables

**Figure 1 ijms-22-00935-f001:**
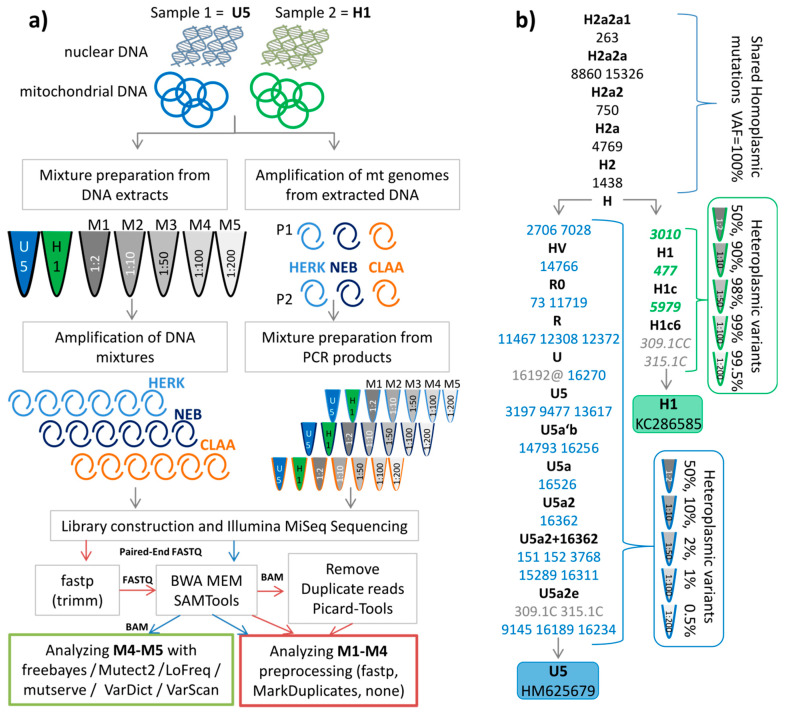
(**a**) Workflow for sample preparation and analysis; on the left side the DNA of two individuals was mixed at five different ratios (M1 1:2, M2 1:10, M3 1:50, M4 1:100, M5 1:200) which then was amplified by three different enzymes (HERK (Herculase), NEB, and CLAA (Clontech)). On the right side, the PCR products, derived from the same DNA, were mixed at the different ratios as described above. (**b**) Phylogenetic tree representing the profiles of two individuals. The sequences are stored in GenBank (accession numbers KC286589 and HM625679). The shared polymorphic positions are indicated in black. Mutations are transitions unless a base is explicitly indicated. Variants in blue denote minor polymorphic sites, green indicates major polymorphic sites expected in the mixtures. Grey positions were not considered in the analysis. The symbol @ indicates that this variant is mutated back to the ancestral allele, hence not expected in the mixture.

**Figure 2 ijms-22-00935-f002:**
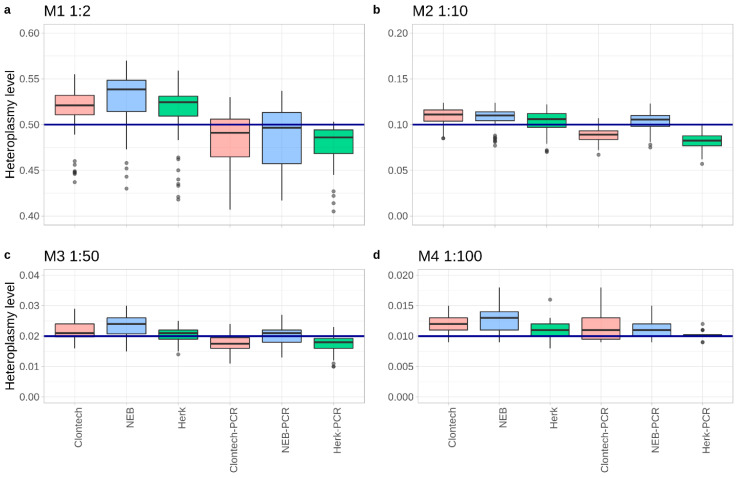
Boxplots showing the averaged mixture level from all experiments down to 1:100 (M4) over all expected positions. The blue line represents the expected heteroplasmy level: (**a**) 0.5 for M1 (1:2), (**b**) 0.1 for M2 (1:10), (**c**) 0.02 for M3 (1:50), and (**d**) 0.01 for M4 (1:100).

**Figure 3 ijms-22-00935-f003:**
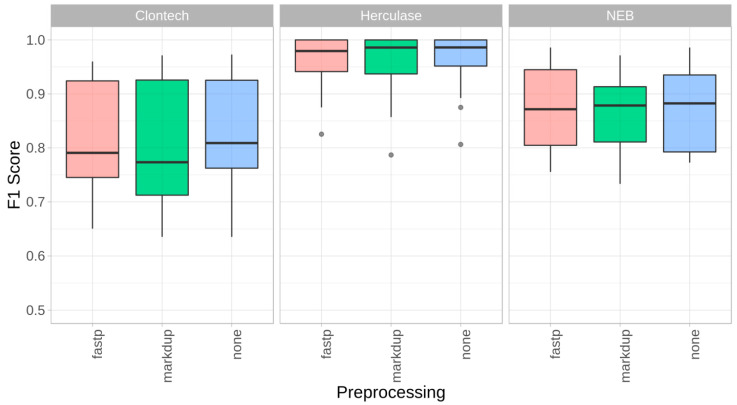
Impact of preprocessing (trimming, PCR duplicate removal). F1 Scores for mixtures M1–M4 for the three polymerases, with fastp’s default trimming + quality filtering (fastp), Picard-Tools MarkDuplicates (markdup), and without preprocessing (none) by using mutserve variant caller, with the default parameters in all three experiments (1% threshold, per base quality Phred score 20, BAQ turned off).

**Figure 4 ijms-22-00935-f004:**
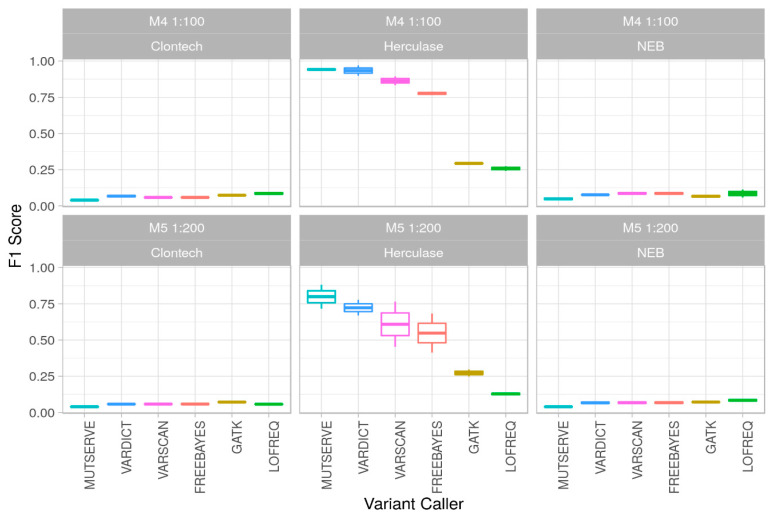
Polymerase and variant calling performance in low-level mixtures. F1 Score (*y*-axis) for mixtures M4 (1:100) and M5 (1:200) for the 3 polymerases (Clontech, Herculase, NEB) for 6 different variant callers (*x*-axis) over two different DNA extraction methods.

**Table 1 ijms-22-00935-t001:** Variant callers and parameters applied for analyzing the mixtures M4 (1:100) and M5 (1:200). Where possible, the threshold for allele frequency was set to 0.4%. Version indicates the applied version. Runtime represents the mean runtime per sample; see Appendix A for further information.

Variant Caller Used in This Project	Parameters	Version	Runtime
mutserve	--level 0.004 --noBaq --baseQ 30	1.3.4	00:35 s
**Variant caller used for comparison**			
freebayes	--min-mapping-quality 30 --min-base-quality 30 --min-alternate-fraction 0.004 --min-alternate-count 5	1.3.1	04:37 s
GATK4 Mutect2	--min-base-quality-score 30 --mitochondria-mode	4.1.8.1	05:16 s
LoFreq	-- B (disable BAQ) --minimum-mapping-quality 30	2.1.5	01:43 s
VarDictJava	-f 0.004 (threshold allele frequency) -q 30 (basequality)	1.7.0	02:41 s
Varscan	--min-var-freq 0.004 --min-avg-qual 30 --min-reads 5 --min-coverage 10	2.4.4	00:44 s

## Data Availability

All 96 Illumina MiSeq paired-end FASTQ files as well as all 48 BAM files processed with BWA-MEM [54] and SAMtools [55] (see Appendix A for details) are available on Zenodo (under https://doi.org/10.5281/zenodo.3991749 and https://doi.org/10.5281/zenodo.4395665, respectively, as well as 4 high-coverage HiSeq BAM files [1] (mixtures 1:2, 1:10, 1:50, and 1:100) https://doi.org/10.5281/zenodo.3339078).

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
