# Peer review of "Analyzing Low-Level mtDNA Heteroplasmy—Pitfalls and Challenges from Bench to Benchmarking"

_ijms, 2021, doi:10.3390/ijms22020935_

Round 1

Reviewer 1 Report

This manuscript and the data obtained by the authors make an important contribution to the standardization of workflow and analysis in studies on detection of heteroplasmic mtDNA variants as well as any other low-level genomic variants using massively parallel sequencing (MPS). The main advantage of this work is the assessment of the impact of pre-analytical factors, preprocessing and use of different variant callers on the MPS-based mtDNA heteroplasmy analysis. 

Manuscript is well-written and brings valuable data worth to be published, however some issues should be clarified and some minor corrections should be made:

  1. Where does the problem of comparing different polymerases when detecting low-level mtDNA variants come from? 
  2. Did the authors check the size and homogeneity of the amplicons obtained by different polymerases using electrophoretic separation? After PCR generation of amplicons and before amplicon fragmentation and subsequent library prep, a visualization in an agarose gel or PAGE should be performed to exclude the presence of non-specific products, especially when comparing different polymerases. I suggest this data should be provided or described in the manuscript.
  3. In this work the authors discovered the emergence of a systematic artifact at position 3210 present in any independent sequencing reaction at equal levels caused exclusively by the NEB polymerase. In a previously published work by the authors (Fendt et al, Cancers 2020) on evaluating mtDNA heteroplasmy in a prospective oral squamous cell carcinoma, this variant was declared as heteroplasmic, and NEB polymerase was used for the amplicon generation. Does this mean that the declared heteroplasmy was an artifact of the NEB polymerase? This issue might be discussed in a manuscript.
  4. In figure 1b the position 16526 appears twice. This is probably a mistake, as according to Phylotree 17, 16256 variant (both with 14793) is a marker of U5a haplogroup, and 16526 variant determines U5a2 haplogroup. Also it is unclear in 16162@ variant what the “@” sign means, in my opinion the description is needed.

Finally, the text should be checked for correctness of grammar, punctuation and abbreviation use, for example:

  • line 54: “mixture carryover)” - extra parenthesis
  • line 85: “where not” - i suppose, “were not”
  • line 113: “false positives variants” - should be changed to “false positives” or “false positive variants”
  • line 162: “fromPCR products” - the space is missing,
  • lines 201-202: “systematic errors … would - if not questioned … - are prone” - please correct or rephrase
  • line 268 - “spectrophotometically” 
  • line 268 - “1:50,1:100” - the space is missing
  • lines 121, 123, 325 - please use text instead of numbers
  • line 116: BAQ first appears as an abbreviation, in my opinion it should be explained

Reviewer 2 Report

The authors provide a study of performance for different PCR enzymes as well as variant callers to detect low level mtDNA variants in artificially created mixtures. It is nice to see a paper dedicated to these essential factors to make the readers aware of the influence of different enzymes and analysis methods. Subjects that are certainly worth investigating.

The manuscript is well-written and some parts are nicely illustrated, but a more in depth discussion and inclusion of a few more detailed results would increase the value for the readers.

  1. While most of the data focusses on MiSeq data, HiSeq data and Sanger data is mentioned as ‘golden standard’ for comparison. It would be worth mentioning that slight differences exist in ‘error profiles’ for different Illumina platforms (Schirmer et al. 2016, BMC BioInformatics).
  2. Could you add a (sup.) table with the observed ‘golden standard’ variants of the samples used for the mixtures. Since you mention the dataset to be available as benchmarking dataset, this information should be added as well.
    1. Did any of these samples contain heteroplasmic variants (if yes, could you include these with the respective percentages)
  3. You are using several variant callers which will not all be familiar to each reader. While you do describe the different callers in the sup. data, a condensed table (or other form of overview) in the main text of the manuscript displaying the main settings and differences (even if default settings, like used threshold for calling minor variants) of the different variant callers can help the reader to interpret the observed differences.
  4. While you interestingly discuss several factors that could influence the quality, and limits of detecting low level variants, it would be worth mentioning which other factors could influence the results.
    1. The cause of the false positives is not discussed while this can be interesting to the readers but can also provide opportunities to improve the analysis.
    2. While some of the variant callers might take this into account, you haven’t mentioned anything about checking for sequence orientation bias (did you look at this for the false positives?). PCR errors are observed in both orientations while sequence errors are often strongly biased for one orientation.
    3. It would also be worth mentioning ways to improve the quality and increase sensitivity (i.e. shorter library fragments would lead to more overlap of the PE reads which can increase quality)

A few small tips:

  1. 162 fromPCR is written as one word
  2. Instead of only using the name M4, M5… I would advise to mention the mixture ratio’s next to the name in each figure. This will aid readability.

Round 2

Reviewer 2 Report

No further suggestions. Everything was incorporated / answered very well.